# Masked Randomized Trial of Epinephrine versus Vasopressin in an Ovine Model of Perinatal Cardiac Arrest

**DOI:** 10.3390/children10020349

**Published:** 2023-02-10

**Authors:** Munmun Rawat, Sylvia Gugino, Carmon Koenigsknecht, Justin Helman, Lori Nielsen, Deepika Sankaran, Jayasree Nair, Praveen Chandrasekharan, Satyan Lakshminrusimha

**Affiliations:** 1Department of Pediatrics, University at Buffalo, Buffalo, NY 14203, USA; 2Department of Pediatrics, UC Davis Medical Center, Sacramento, CA 95817, USA

**Keywords:** resuscitation, vasopressin, epinephrine

## Abstract

Background: Current neonatal resuscitation guidelines recommend the use of epinephrine for bradycardia/arrest not responding to ventilation and chest compressions. Vasopressin is a systemic vasoconstrictor and is more effective than epinephrine in postnatal piglets with cardiac arrest. There are no studies comparing vasopressin with epinephrine in newly born animal models with cardiac arrest induced by umbilical cord occlusion. **Objective:** To compare the effect of epinephrine and vasopressin on the incidence and time to return of spontaneous circulation (ROSC), hemodynamics, plasma drug levels, and vasoreactivity in perinatal cardiac arrest. **Design/Methods:** Twenty-seven term fetal lambs in cardiac arrest induced by cord occlusion were instrumented and resuscitated following randomization to epinephrine or vasopressin through a low umbilical venous catheter. **Results:** Eight lambs achieved ROSC prior to medication. Epinephrine achieved ROSC in 7/10 lambs by 8 ± 2 min. Vasopressin achieved ROSC in 3/9 lambs by 13 ± 6 min. Plasma vasopressin levels in nonresponders were much lower than responders after the first dose. Vasopressin caused in vivo increased pulmonary blood flow and in vitro coronary vasoconstriction. **Conclusions:** Vasopressin resulted in lower incidence and longer time to ROSC compared to epinephrine in a perinatal model of cardiac arrest supporting the current recommendations for exclusive use of epinephrine in neonatal resuscitation.

## 1. Introduction

Perinatal asphyxia is a leading cause of neonatal mortality and morbidity throughout the world [1,2,3]. Asphyxiated newborns who fail to respond to optimized positive pressure ventilation (PPV) and chest compressions have high mortality with likelihood of neurodevelopmental disability among survivors [4,5]. Epinephrine is recommended by the American Heart Association (AHA), American Academy of Pediatrics (AAP) Neonatal Resuscitation Program (NRP), and International Liaison Committee on Resuscitation (ILCOR) if the baby’s heart rate remains <60/min after at least 30 s of effective PPV and another 60 s of chest compressions coordinated with PPV with 100% oxygen [6]. During cardiopulmonary resuscitation (CPR), epinephrine improves systemic circulation by inducing systemic vasoconstriction while cardiac compression generates forward blood flow to perfuse the heart, lungs, and brain. Providing chest compressions without delivering an optimal dose of epinephrine does not adequately increase carotid flow [7]. However, epinephrine can cause high blood pressure, rapid heart rate, increased myocardial oxygen consumption, and postarrest arrhythmias and can lead to cardiac dysfunction (secondary to beta-1 adrenergic effects) [8]. During experimental cardiopulmonary resuscitation, high-dose epinephrine appears to induce vasoconstriction of cortical cerebral blood vessels resulting in redistribution of blood flow from the superficial cortex [9].

Vasopressin has been identified as a promising vasopressor during cardiac arrest in animal models [10] and clinical studies in adults [11]. It has been shown that vasopressin improves vital organ blood flow during prolonged CPR and pulseless electrical activity [12] and optimizes blood flow in hypovolemic shock [13]. Vasopressin exerts its systemic vasoconstrictor and possibly a pulmonary vasodilator effect through V1 receptors without a significant direct effect on the heart [14]. McNamara et. al. showed improved survival, lower postresuscitation troponin, and less hemodynamic compromise after cardiac arrest in newborn piglets with vasopressin as compared to epinephrine during resuscitation [15]. Vasopressin may be uniquely suited for neonatal asphyxia characterized by respiratory asphyxia leading to arrest with high pulmonary vascular resistance (PVR) due to its pulmonary vasodilatory properties and intense systemic vasoconstriction through nonadrenergic mechanisms without any effect on the heart [16]. Additionally, vasopressin has shown to increase cerebral blood flow during hypoxemia by selective dilatation of cerebral vessels [17] (Eisenach, Tong et al. 1992).

Our objectives were to compare the effect of epinephrine and vasopressin in a fetal lamb model of cardiac arrest induced by umbilical cord occlusion on the incidence and time to return of spontaneous circulation (ROSC), systemic and pulmonary hemodynamics, plasma drug levels, and in-vitro vascular reactivity.

## 2. Materials and Methods

This study was approved by the Institutional Animal Care and Use Committee at the State University of New York at Buffalo. Time-dated term (139–141-day gestation) pregnant ewes were obtained from May Family Enterprises (Buffalo Mills, PA, USA), anesthetized by intravenous diazepam and ketamine, intubated with a 10.0 mm-cuffed endotracheal tube (ETT) and ventilated with 21% oxygen and 2–3% isoflurane at 16 breaths/min. The ewes were continuously monitored with a pulse oximeter and an end-tidal CO_2_ (EtCO_2_) monitor. Following cesarean section, fetal lambs were partially exteriorized and intubated with a 4.5 mm cuffed ETT as previously demonstrated [18]. The fetal lung fluid in the ETT was drained passively by gravity by tilting the head to the side. Subsequently, the ETT was occluded to prevent gas exchange during gasping in the asphyxial period. Catheters were inserted into the jugular vein (for fluid and medication administration) and right carotid artery (for blood sampling). A 2 mm flow probe (Transonic Systems Inc, Ithaca, NY, USA) was placed around the left carotid artery. A left thoracotomy was performed, and a 4 mm flow probe was placed around the left pulmonary artery. The thoracotomy was closed in layers. Electrocardiogram (ECG) leads were attached at the right axilla, left axilla, and right inguinal area (three-lead ECG). The ECG100C (Biopac Systems, Inc., Goleta, CA, USA) was used with Acknowledge Software to record tracings of leads I, II, and III. Following instrumentation, the umbilical cord was occluded and then cut, and the lambs were moved from the maternal abdomen to the radiant warmer. During the asphyxial period (prior to resuscitation), an umbilical arterial catheter was inserted to measure continuous invasive blood pressures. A low umbilical venous catheter (2–3 cm below the skin) was placed for epinephrine or vasopressin administration.

As shown in the flow chart (Figure 1), a five-minute period of cardiac arrest was observed prior to initiating resuscitation to minimize chances of ROSC with PPV alone. Cardiac arrest was defined by the absence of carotid blood flow, arterial blood pressure, and audible heart rate. Resuscitation began by providing PPV with 21% O_2_ by means of a T-piece resuscitator at a rate of 40 breaths/min and initial pressures of 35/5 cm H_2_O. Peak inspiratory pressure (PIP) was adjusted as needed to obtain adequate chest rise. Chest compressions were initiated after 30 s of effective ventilation. Upon initiation of chest compressions, inspired O_2_ was increased to 100% as per current NRP recommendations. The lambs were randomized to epinephrine or vasopressin group using sealed opaque envelopes. An additional lab personnel who was not involved in the resuscitation prepared the medications in a separate room. The resuscitators were blinded as both the medications were diluted to 1 mL. The first dose of drug, epinephrine (0.03 mg/kg) or vasopressin (0.4 U/kg), was administered into the low umbilical venous catheter if return of spontaneous circulation (ROSC) had not been achieved after 5 min of effective PPV and CC. A 1cc normal saline flush was used after every medication push. This was the same throughout the protocol in both the groups. Repeated doses of epinephrine/vasopressin were administered every 3 min for a maximum of 4 doses or until ROSC. We defined ROSC as a sustained heart rate >100/min with a systolic blood pressure of >30 mmHg. Resuscitation was stopped at 20 min if there was no ROSC.

Following ROSC, the lambs were placed on a ventilator. The PIP and rate were adjusted gradually based on tidal volumes (goal 8–9 mL/kg) and PaCO_2_, and the FIO_2_ was adjusted by 5–10% every 30 s to maintain preductal saturations between 85–95%. Hemodynamic parameters were continuously monitored. Lambs were euthanized by administering 100 mg/kg pentobarbital sodium (Fatal-Plus Solution; Vortech Pharmaceuticals, Dearborn, MI, USA) approximately 20 min into the study.

In-Vitro Vessel Bath Study: Coronary, carotid, pulmonary, and femoral vessel rings were obtained from a different set of asphyxiated lambs, and vasoconstrictor or dilator response to a concentration equivalent to circulating levels following administration of clinical doses of 0.4 IU/kg vasopressin or 0.03 mg/kg epinephrine were studied in a vessel bath. Changes in vascular tone were recorded as gram of tension/gram of tissue weight (g/g).

Based on pilot data, we needed 9 lambs in each group to detect significant differences in incidence of ROSC, plasma epinephrine levels, and adverse events including hypertension and tachycardia with a power of 80% and a type 1 probability of 0.05.

Plasma drug levels (epinephrine and vasopressin) were measured at various intervals using commercially available ELISA kits (Epinephrine-Eagle Biosciences, Nashua, NY, USA and Vasopressin—Cayman Chemical, Ann Arbor, MI, USA). Arterial blood flow and pressures were continuously recorded using AcqKnowledge Acquisition & Analysis Software (BIOPAC systems, Goleta, CA, USA). The values were averaged over 1 min and extracted from BIOPAC system. Continuous variables are expressed as mean and standard deviation. Categorical variables were analyzed using chi square test or Fisher’s exact test as appropriate. Continuous variables were analyzed by one-way ANOVA between groups with Fisher’s post hoc test within groups. Statview 4.0 software was used. Statistical significance was defined as *p* < 0.05.

## 3. Results

Twenty-seven lambs were asphyxiated to cardiac arrest by umbilical cord occlusion. Eight lambs achieved ROSC with PPV and chest compressions prior to administration of medication.

Baseline characteristics of all lambs are shown in Table 1. The epinephrine and vasopressin groups were similar in birth weight, sex distribution, gestational age, multiplicity, arterial pH, PaO_2_, PCO_2_, and lactate physiological parameters such as blood pressure, heart rate, pulmonary artery blood flow, and carotid artery blood flow prior to the start of the experimental protocol. Time to asystole was similar in both the groups.

### 3.1. Return of Spontaneous Circulation

In the epinephrine group, seven out of 10 lambs achieved ROSC with time to ROSC 8 ± 2 min, whereas with vasopressin, three out of nine lambs achieved ROSC in 13 ± 6 min. The number of doses required to achieve ROSC in the epinephrine group was smaller (1.2 ± 0.5) as compared to the vasopressin group (3 ± 1.4). Five out of ten (50%) lambs achieved ROSC after the first dose of epinephrine, whereas only one out of nine lambs (11%) achieved ROSC after the first dose of vasopressin.

### 3.2. Systemic Hemodynamics

The mean carotid artery blood flow for 20 min after ROSC was similar between epinephrine (23 ± 3.8 mL/kg/min) and vasopressin groups (23 ± 3.3 mL/kg/min; *p* = 0.39). There was no difference in the systolic blood pressures between epinephrine (65 ± 26 mmHg) and vasopressin (66 ± 19 mmHg, *p* = 0.88) group after ROSC. The diastolic pressures were also similar (48 ± 19 mmHg in epinephrine and 50 ± 15 mmHg in vasopressin group, *p* = 0.38). After achieving ROSC, the lambs that received epinephrine had higher heart rates (205 ± 29 bpm) as compared to those that received vasopressin (178 ± 44 bpm, *p* = 0.001). See Figure 2.

### 3.3. Carotid and Pulmonary Hemodynamics

The box and whisker plot (Figure 3A) shows no difference in the carotid artery blood flow in the lambs that received epinephrine 23 (20–26) mL/kg/min versus vasopressin 24 (23–26) mL/kg/min. In contrast, vasopressin resulted in increased pulmonary blood flow 75 (59–83) mL/kg/min after ROSC when compared to epinephrine 38 (33–43) mL/kg/min expressed as median (1st and 3rd interquartile range) and a *p* value of 0.0001. See Figure 3B.

### 3.4. Plasma Drug Level

After the first dose of epinephrine, the lambs that achieved ROSC (responders) had similar plasma epinephrine levels 768 ± 720 ng/mL) as compared to the lambs that did not achieve ROSC (nonresponders, 741 ± 474 ng/mL). However, responders in the vasopressin group had higher plasma vasopressin levels (68 ± 26 ng/mL) as compared to nonresponders (43 ± 26 ng/mL) after the first dose of vasopressin. See Figure 4.

### 3.5. In-Vitro Vessel Study

Coronary, carotid, pulmonary, and femoral vessel rings were obtained from a different set of asphyxiated lambs, and vasoconstrictor or dilator response to a concentration equivalent to circulating levels following administration of clinical doses of 0.4 IU/kg vasopressin or 0.03 mg/kg epinephrine were studied in a vessel bath. Changes in vascular tone were recorded as gram of tension/gram of tissue weight (g/g). We observed increased coronary artery relaxation with epinephrine (minus 49 ± 38 g/g) as compared to vasopressin (+43 ± 63 g/g, *p* < 0.01). There was an increased tendency of carotid artery constriction with epinephrine (254 ± 202 g/g) as compared to vasopressin (43 ± 55 g/g, *p* = 0.06). There was an increased tendency to constrict the pulmonary arterial rings with epinephrine (44 ± 7 g/g) as compared to vasopressin (17 ± 35 g/g, *p* = 0.12). No difference was seen in femoral arterial ring constriction (467 ± 591 g/g with epinephrine as compared to 400 ± 418 g/g with vasopressin, *p* = 0.9). See Figure 5.

## 4. Discussion

We conducted a head-to-head comparison of epinephrine and vasopressin in a transitional lamb model of severe asphyxia leading to perinatal cardiac arrest. Contrary to a previous study in postnatal piglet model, incidence of ROSC with epinephrine was more common as compared to vasopressin (70% versus 33% respectively). Additionally, ROSC achieved by epinephrine was quicker (8 ± 2 min vs. 13 ± 6 min) and required fewer doses of medication (1.2 ± 0.5 vs. 3 ± 1.4) compared to vasopressin. This is the only blinded, randomized comparison of vasopressin with epinephrine in the setting of neonatal resuscitation in a transitional model of perinatal cardiac arrest induced by umbilical cord compression.

Epinephrine is a catecholamine with inotropic (increases cardiac contractility), lusitropic (myocardial relaxation), chronotropic (increases heart rate), and vasoconstrictor properties. The vasoconstrictor properties are mediated by α adrenergic receptors and are primarily responsible for the effectiveness of epinephrine during CPR in adults and postnatal children [19,20,21]. Administration of epinephrine is believed to induce peripheral vasoconstriction, which results in elevated systemic vascular resistance and an increase in coronary perfusion pressure to improve coronary blood flow [22]. The significance and safety of the β-adrenergic effects of epinephrine are controversial because they may increase myocardial work and oxygen demand [23]. The beneficial effects on survival and myocardial function may be age- and species-specific due to variability in density and binding of receptors or other pharmacokinetic factors. We speculate that human neonates, lambs, and piglets have different distribution of α and β-adrenergic receptors leading to species differences [15].

Vasopressin (or antidiuretic hormone) is an intense systemic vasoconstrictor [24] and pulmonary vasodilator in adults [16]. Vasopressin has been shown to improve total cerebral blood flow and oxygenation during cardiopulmonary resuscitation [25]. Few adverse effects such as pulmonary edema have been reported in adults [26]. Vasopressin is more effective than catecholamines in inducing in vitro vasoconstriction in the presence of acidosis [27]. These advantages might partly explain the higher efficacy of vasopressin compared to epinephrine in adult CPR for asystole [28,29]. In our study, we observed higher pulmonary blood flow with vasopressin, which may be attributed to the pulmonary vasodilator properties of vasopressin or intense systemic vasoconstriction resulting in left-to-right shunt across the ductus arteriosus. However, in our in vitro vessel bath study, the pulmonary artery did not relax with vasopressin.

We believe the advantages of vasopressin over epinephrine observed in postnatal piglets in the study by McNamara (McNamara, Engelberts et al. 2014) as well as in human adults are related to a more powerful systemic vasoconstrictor effect associated with pulmonary vasodilation. However, some of these advantages may be related to the postnatal nature of the model (air-filled lungs with low pulmonary vascular resistance and closed or narrowing ductus arteriosus) [30]. In lamb models of perinatal asphyxia with an open ductus, chest compressions and epinephrine do not markedly elevate diastolic pressure [31]. Hence, a powerful systemic vasoconstrictor such as vasopressin is probably not as effective as epinephrine that has both α-adrenergic effects on systemic circulation combined with chronotropic β-adrenergic effect on the heart. The higher heart rate during the post-ROSC phase in lambs that received epinephrine is probably secondary to this β-effect. Higher pulmonary blood flow and venous return to the left heart with vasopressin might account for higher preload and lower heart rate in this group compared to epinephrine. Although we did not observe a difference in carotid flow per minute between the two groups, stroke carotid flow volume (carotid flow per heartbeat) was higher with vasopressin (0.175 ± 0.08 mL/kg) compared to epinephrine (0.109 ± 0.02 mL/kg, *p* < 0.001).

In addition, it has been shown that the pulmonary vasodilator effect of vasopressin is decreased in newborn rodent models compared to adults [32]. There is also a concern for tissue hypoperfusion (mainly splanchnic) in extremely low-birth-weight infants with vasopressin [33]. Our isolated vessel reactivity studies showed higher tendency of carotid vasoconstriction with epinephrine with significant coronary vasodilation. In contrast, vasopressin induced intense coronary vasoconstriction. We speculate that vasopressin resulted in lower and slower incidence of ROSC most likely secondary to the constricting effect on coronaries limiting coronary perfusion, inability to achieve higher diastolic blood pressures due to the presence of the ductus, and to achieving lower plasma levels among nonresponders.

We used a dose of 0.4 U/Kg vasopressin for our study. This dose was extrapolated from adult literature and was further supported by dose–response resuscitation studies in postnatal piglets [15]. We did not see any adverse effects secondary to vasopressin. Based on our study, the animals that achieved ROSC had higher vasopressin levels in the plasma as compared to ‘no ROSC’. We speculate two potential mechanisms for low vasopressin levels. The first is inadequate drug delivery—this mechanism is unlikely as it was directly delivered into the umbilical venous circulation. A different volume or path of distribution through the hepatic venous system might have reduced plasma levels. Finally, the use of 1 mL of flush was standard as per NRP at the time of this study. However, a higher volume of flush is recommended by NRP, and use of 3 mL of flush might have potentially enhanced vasopressin levels [34]. Hence, we speculate that a higher dose of vasopressin (0.6–0.8 U/kg) or a higher dose of flush should be studied in this model.

Vasopressin was taken out of adult resuscitation guidelines in 2015 as it offers no advantage as a substitute for epinephrine in cardiac arrest [35]. This was based on a single randomized control trial (RCT) enrolling 336 patients that compared multiple doses of standard-dose epinephrine with multiple doses of standard-dose vasopressin in cardiac arrest and showed no benefit with the use of vasopressin for ROSC or survival to discharge with or without good neurologic outcome [36].

The strengths of our study include masking of providers and randomization prior to delivery. Fetal lambs are similar in size to the human neonate, and they have comparable pulmonary physiology. The fetal lamb is an accepted standard for fetal and newborn physiology and pathophysiology. We administered epinephrine and vasopressin through the low umbilical venous catheter similar to human neonates in the delivery room. Instrumentation and vasoreactivity studies gave us added information about systemic and pulmonary hemodynamics and vasoreactivity.

## 5. Conclusions

Given the unpredictable occurrence of birth asphyxia and ethical, moral, and practical constraints of obtaining consent and plasma levels in the delivery room, a randomized clinical study evaluating these medications is unlikely to be conducted. The current neonatal resuscitation guidelines for medication use in the delivery room are predominantly based on animal studies. The results of our study do not support addition of vasopressin to the current NRP algorithm and agree with current guidelines to exclusively use epinephrine for bradycardia not responding to PPV and chest compressions.

## Figures and Tables

**Figure 1 children-10-00349-f001:**
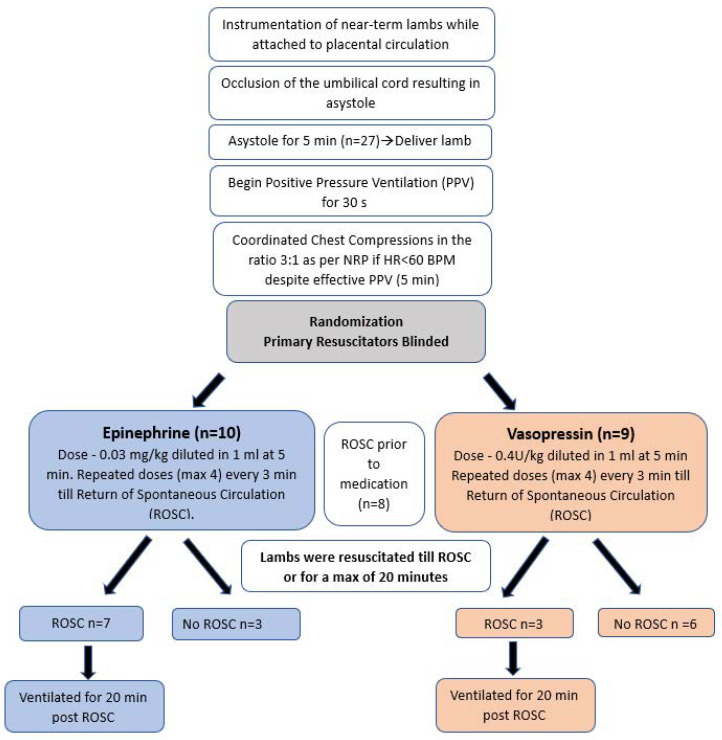
Flow chart showing the experimental design. Near-term lambs were instrumented, asphyxiated to asystole, and resuscitated as per NRP guidelines after 5 min of asystole. In this blinded study, lambs were randomized to epinephrine and vasopressin group. The first dose of medication was given via low-lying umbilical venous catheter after 5 min of positive pressure ventilation and chest compressions.

**Figure 2 children-10-00349-f002:**
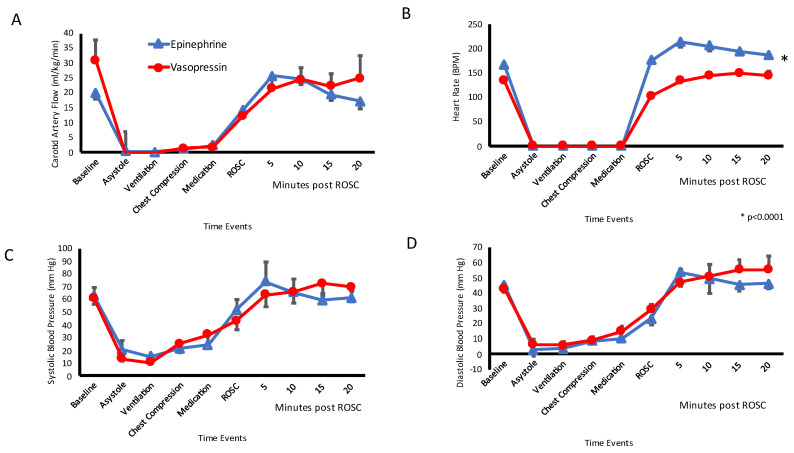
Systemic Hemodynamics: The following line graphs show hemodynamic parameter on the y-axis and the events on the x-axis. The blue open triangles represent the epinephrine group and the red circles represent the vasopressin group. (**A**) The mean carotid blood low (CBF) (obtained from the left carotid artery) in ml/kg/min is shown on the y-axis. The CBF was not different between the two groups. Data is represented as average and standard error of mean. (**B**) Heart rate (Beats per minute) as measured by EKG was higher in the epinephrine group as compared to vasopressin group after achieving return of spontaneous circulation (ROSC). (**C**) Systolic Blood Pressure in mm Hg measured invasively via pressure probe in ascending aorta were similar in both the groups. (**D**) Diastolic Blood Pressure in mm Hg were similar in both the groups.

**Figure 3 children-10-00349-f003:**
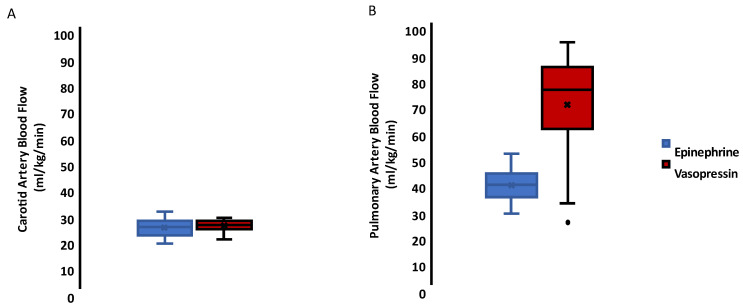
Hemodynamics: Box-and-whisker plot of the difference in measured blood flow in the lambs that received epinephrine (blue) and those that received vasopressin (red) groups. Boxes are bound by 1st and 3rd quartiles, with the center line indicating the median. Error bars represent the range of the data, and dots signify outliers and crosses the means. (**A**) There was no difference in carotid blood flow between epinephrine and vasopressin groups. (**B**) Administration of vasopressin resulted in increased pulmonary blood flow after ROSC when compared to epinephrine (*p* = 0.0001).

**Figure 4 children-10-00349-f004:**
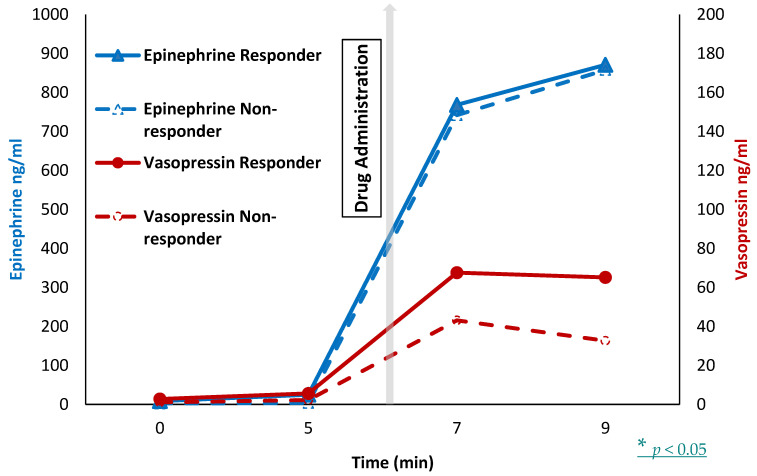
Plasma drug level: The graph shows plasma drug levels after the first dose of drug administration. The plasma epinephrine levels are shown on the primary y axis and the plasma vasopressin levels on the secondary y axis. The x axis shows time in minutes from onset of resuscitation. The vertical grey bar shows the time of first drug (epinephrine or vasopressin) administration. The blue solid triangles represent the lambs that achieved return of spontaneous circulation (ROSC) after epinephrine (responders), and blue open triangles represent the lambs that did not achieve ROSC after epinephrine (nonresponders). The red solid circles represent the lambs that achieved ROSC after vasopressin (responders), and red open circles represent the lambs that did not achieve ROSC after vasopressin (nonresponders). After the first dose of epinephrine, the responders had similar plasma epinephrine levels (768 ± 720 ng/mL) as compared to the nonresponders (741 ± 474 ng/mL). However, responders in the vasopressin group had higher plasma vasopressin levels (68 ± 26 ng/mL) as compared to nonresponders (43 ± 26 ng/mL) after the first dose of vasopressin (* *p* < 0.05).

**Figure 5 children-10-00349-f005:**
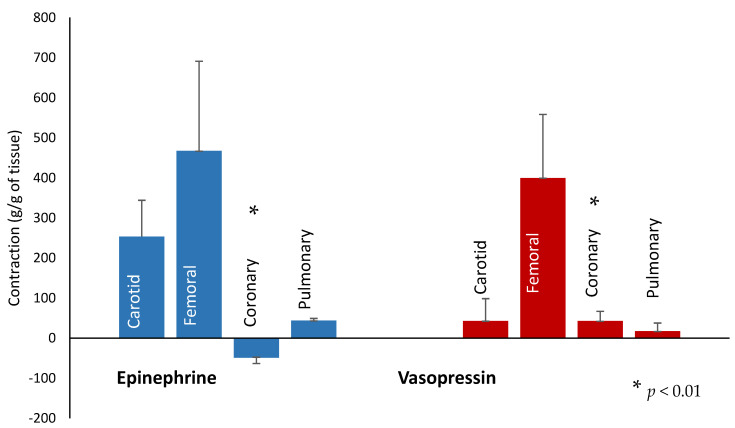
In vitro vessel study: The bar graph shows response of coronary, carotid, pulmonary, and femoral vessel rings (x axis) obtained from a different set of asphyxiated lambs to a concentration equivalent to circulating clinical doses (0.4 IU/kg vasopressin or 0.03 mg/kg epinephrine) of vasopressin and epinephrine in a vessel bath recorded as gram/gram (g/g) on y axis. Blue bars represent the vessels exposed to in vitro epinephrine, and red bars in vitro vasopressin. We observed increased coronary artery relaxation with epinephrine as compared to vasopressin (* *p* < 0.01).

**Table 1 children-10-00349-t001:** Baseline Characteristics.

Parameter	Epinephrine (*n* = 10)	Vasopressin (*n* = 9)
Weight (Kg)	4.3 ± 1.2	4.3 ± 0.7
Sex	5 F, 5 M	7 F, 2 M
Baseline pH	7.21 ± 0.24	7.24 ± 0.06
Baseline PaCO_2_ (mm Hg)	66 ± 21	63 ± 2
Baseline PaO_2_ (mm Hg)	20 ± 3	22 ± 6
Baseline Lactate (mM/L)	3 ± 1	3 ± 1
Baseline CA Flow (mL/kg/min)	24 ± 5	29 ± 12
Baseline PA Flow (mL/kg/min)	19 ± 18	19 ± 15
Time to Asystole (min)	14 ± 6	16 ± 5
Arterial pH at asystole	6.88 ± 0.06	6.86 ± 0.05
PaCO_2_ (mm Hg) at asystole	139 ± 16	135 ± 14
PaO_2_ (mm Hg) at asystole	6 ± 3	7 ± 3
Arterial Lactate (mM/L) at asystole	9 ± 1.5	9 ± 2

## Data Availability

Data presented in this manuscript will be available after complete analysis (2 h data) is performed on request.

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
