# Peer review of "Masked Randomized Trial of Epinephrine versus Vasopressin in an Ovine Model of Perinatal Cardiac Arrest"

_children, 2023, doi:10.3390/children10020349_

Round 1
Reviewer 1 Report
Dear author,
Your article Masked Randomized Trial of Epinephrine versus Vasopressin 2 in an Ovine Model of Cardiac Arrest should be improve
- Please provide information about disign of trial (flow-chatt)
- iHemodynamic parameters were continuously monitored (please provide dosage)
- Hemodynamic parameters were continuously monitored - provide how (device and others)
During the asphyxial period (prior to resuscitation - mean time of period
Rivise please.
Author Response
Dear Editor and Reviewer,
Thank you for considering our submission for publication. We have revised the manuscript to address all the concerns of the reviewers. The response to reviewers to individual points is included in the reviewer comments attached (in blue italics). We thank the reviewers for their detailed review of the manuscript. We feel that these revisions have improved the clarity and readability of this manuscript. A revised manuscript with tracked changes is uploaded.
Sincerely,
Munmun

Reviewer 2 Report
Summary:
In this ovine model of cardiac arrest, the authors examined the effects of vasopressin and epinephrine being administered in a masked and randomized fashion. They found that vasopressin results in a lower and longer time to return of spontaneous circulation and concluded that this model supported the current recommendations of use of epinephrine in newborn resuscitation.
Title and abstract:
Appropriate and succinct. Well written abstract.
Introduction:
1. Line 32 – consider changing birth asphyxia to perinatal asphyxia
2. Nicely written introduction giving the readers a good overview of the previous work on the topic and why the authors want to pursue this study.
3. It would be good to mention the effects of vasopressin on cerebral circulation.
Methods:
1. Line 99 – please elaborate further on the medication/medication administration since the title states “masked randomized trial…”. Were the medications (epinephrine and vasopressin) pre-made? If so, was there any difference in the package/ presentation? Who had access to these?
2. Line 100 – who performed the dilution of the medication to 1 ml and was this done in the vicinity of the investigators. The authors need to detail further about the medication/ preparations to ensure that the readers are convinced that the investigators were truly blinded and therefore not biased.
3. Was there a difference in the strength of the dilution for both the medications?
4. Was a flush used following administration of the medication? If so, what was used and what volume? Was this volume constant for all the doses in the 2 groups?
Results:
1. Was cardiac output measured?
2. What proportion of the lambs in the vasopressin group had high plasma levels?
3. Figure 2b – would be useful for the readers to have pulmonary blood flow in lambs with high and low plasma vasopressin levels.
Discussion:
1. Line 251; please re-phrase this sentence as it is not clear what the authors are trying to convey.
2. Line 257; the authors propose that the combined beta and alpha activity of epinephrine as being a reason for higher heart rate through predominantly beta activity. Could it be possible that the vasopressin group had improved pulmonary blood flow resulting in a higher preload and improved cardiac output and therefore a lower heart rate? Data on cardiac output if available would be very useful.
3. Line 266: Slower incidence of ROSC in the vasopressin group was thought to be due to limited coronary perfusion, low diastolic BP and lower drug levels. Since this is the only study examining these 2 medications as stated by the authors, it would be very useful to compare the 2 subgroups (of lambs with higher plasma vasopressin vs lower plasma vasopressin level) for the outcome of ROSC.
4. Is it possible that vasopressin administration resulted in over expression of vasopressin receptors which reduced beta receptor responsiveness as demonstrated in other animal models (Tilly ea 2014, Circulation) resulting in lower heart rate in this group? It would be useful to see if there was a difference in the heart rate among the high and low drug level groups.
5. Line 273; is there a basis for this recommendation of a higher dose of vasopressin. Based on the discussion and results, the readers may be dissuaded from using a higher dose given the effects of a lower dose vasopressin especially if there are concerns with coronary perfusion. Again, providing physiological data from lambs with higher and lower vasopressin levels may further support this recommendation by the authors.
6. How might the authors explain the difference for the high and low plasma vasopressin levels?
7. Was there any evidence to suggest that the medication could have leaked through a low lying UVC? If not, this should be explicitly stated so that the readers are clear about this and that this was not a potential reason for low plasma levels of vasopressin.
References:
1. Reference number 3 – please re-check this as the entered address does not seem to work.
I want to thank the authors for carrying out this very important work in this challenging area which is often overlooked. I strongly believe that the manuscript in its current form has not reached its full potential and several avenues can be explored further which will provide investigators with very useful physiological data for future work.
Author Response
Dear Editor and Reviewers,
Thank you for considering our submission for publication. We have revised the manuscript to address all the concerns of the reviewers. The response to reviewers to individual points is included in the reviewer comments attached (in blue italics). We thank the reviewers for their detailed review of the manuscript. We feel that these revisions have improved the clarity and readability of this manuscript. A revised manuscript with tracked changes is uploaded.
SIncerely,
Munmun

Reviewer 3 Report
This is a very good manuscript. I only have a few suggestions in order to improve it:
1. I think the title would be improved if it included the exact category of subjects, so something along the lines of “[...] an Ovine Model of Fetal/Neonatal/Perinatal Cardiac Arrest” would be a tremendous help for the reader.
2. In the results section (rows 163-173) there is mention of a different dataset used for separate in vitro findings. I feel the respective results, as well as Figure 4, do not belong here and are forced somehow in this context, especially since nothing about them is mentioned in the Discussions section. A different dataset equals a different manuscript, if the authors wish to show their findings.
3. I feel Figure 1 would be improved if the graphs for systolic BP and diastolic BP would be either side-by-side or one on top of the other. It just doesn’t compute in my head to have them diagonally.
4. The authors should pay closer attention to the way the references are written.
Author Response
Dear Editor and Reviewers,
Thank you for considering our submission for publication. We have revised the manuscript to address all the concerns of the reviewers. The response to reviewers to individual points is included in the reviewer comments below (in blue italics). We thank the reviewers for their detailed review of the manuscript. We feel that these revisions have improved the clarity and readability of this manuscript. A revised manuscript with tracked changes is uploaded.
We believe that the changes suggested by the reviewer have helped us strengthen our manuscript to the full potential. We again thank the reviewer for reviewing our manuscript and for their valuable feedback.
Sincerely,
Munmun Rawat
